# Process Optimization of Palm Oil Mill Effluent-Based Biosurfactant of *Halomonas meridiana* BK-AB4 Originated from Bledug Kuwu Mud Volcano in Central Java for Microbial Enhanced Oil Recovery

**Cut Nanda Sari [1], Rukman Hertadi [2], Andre Fahriz Perdana Harahap [1]**, **Muhammad Yusuf Arya Ramadhan [1] and Misri Gozan [1,*]**

[1] Bioprocess Engineering Program, Department of Chemical Engineering, Faculty of Engineering, Universitas Indonesia, Depok 16424, Indonesia; cutnandasari.lemigas@gmail.com (C.N.S.); andrefahriz25@gmail.com (A.F.P.H.); yra.ramadhan@gmail.com (M.Y.A.R.)

[2] Department of Chemistry, Institut Teknologi Bandung, Bandung 40132, Indonesia; rukman@chem.itb.ac.id

* Correspondence: mgozan@che.ui.ac.id

**Abstract:** Biosurfactants are one of the microbial bioproducts that are in most demand from microbial-enhanced oil recovery (MEOR). The production of biosurfactant is still a relatively high cost. Therefore, this study aims to reduce production costs by utilizing palm oil mill effluent (POME) as the main carbon source. This work examines the optimal conditions of biosurfactant production by *Halomonas meridiana* BK-AB4 isolated from the Bledug Kuwu mud volcano in Central Java Indonesia and studies it for EOR applications. The biosurfactant production stage was optimized by varying POME concentration, incubation time, NaCl concentration, and pH to obtain the maximum oil displacement area (ODA) values. A response surface methodology (RSM) and a central composite design (CCD) were used to identify the influence of each variable and to trace the relationship between variables. Optimum biosurfactant production was found at a POME concentration ($v/v$) of 16%, incubation (h) of 112, NaCl concentration ($w/v$) of 4.7%, pH of 6.5, with an oil displacement area of 3.642 cm. The LC-MS and FTIR analysis revealed the functional groups of carboxylic acid or esters, which indicated that the biosurfactant produced belonged to the fatty acid class. The lowest IFT value was obtained at the second and seventh-day observations at a concentration of 500 mg/L, i.e., 0.03 mN/m and 0.06 mN/m. The critical micelle concentration (CMC) of biosurfactant was about 350 mg/L with a surface tension value of about 54.16 mN/m. The highest emulsification activity ($E_{24} = 76\%$) in light crude oil (naphthenic–naphthenic) and could reduce the interfacial tension between oil and water up to 0.18 mN/m. The imbibition experiment with biosurfactant results in 23.89% additional oil recovery for 60 h of observation, with the highest increase in oil recovery occurring at the 18th hour, which is 2.72%. Therefore, this bacterium and its biosurfactant show potential, and the bacterium are suitable for use in MEOR applications.

**Keywords:** biosurfactant; *Halomonas meridiana* BK-AB4; palm oil mill effluent; response surface methodology

## 1. Introduction

One technological development to increase crude oil production is enhanced oil recovery (EOR), which involves using chemicals to facilitate the extraction of crude oil from old wells. EOR technology generally utilizes surfactants, emulsifiers, polymers, and solvents, all of which pose a relatively high risk of environmental pollution [1]. In contrast, microbial enhanced oil recovery (MEOR) technology

exploits microorganisms that produce biosurfactants [2,3]. These biosurfactants can be synthesized from inexpensive substrates, and they are non-toxic and can be degraded naturally, thereby making them an attractive alternative to EOR based on chemicals. Furthermore, the cost of MEOR does not depend on fluctuations in world crude oil prices [4].

One challenge in the application of MEOR is that conditions in petroleum reservoirs are relatively extreme, with temperatures generally higher than 80 °C [5]. Besides, deep reservoirs contain brine (i.e., water with a high dissolved salt content), which provides inhospitable conditions for many organisms [6]. Thus, MEOR requires microorganisms that can survive at high temperatures and salinity and produce biosurfactants with good emulsification activity.

Halophilic bacteria can live in environments with high salinity due to their ability to balance osmotic pressure in the environment and their resistance to the effects of denaturation of salt. Halophilic bacteria can grow at salt concentrations ≥ 150 g/L [7]. *Halomonas meridiana* is a halophilic bacterium that can tolerate a wide range of salinities and temperatures [8]. *H. meridiana* BK-AB4, which can produce extracellular lipases, is endemic in Indonesia and was successfully isolated from the Bledug Kuwu mud volcano in the Grobogan district, Central Java [9]. Mud volcanoes like Bledug Kuwu have temperatures ranging from 2 °C to 100 °C [10]. Thus, the conditions in mud volcanoes are like those found in oil wells, and they provide an optimum environment for the identification of bacterial strains capable of surviving in oil wells.

Previous research examined the ability of *H. meridiana* BK-AB4 to produce biosurfactants using olive oil and palm oil substrate [3]. However, as these two vegetable oils are food commodities with high demand, their use in biosurfactant production would give rise to competition with other sectors, especially the food sector. Biosurfactant synthesis by microorganisms occurs when they are cultivated in lipid-rich media [3].

In 2014, Indonesia produced 33 million tons of palm oil and three times more liquid waste [11]. Palm oil mill effluent (POME) or palm oil wastewater contains up to 4000 mg/L of lipids, as well as carbohydrates, proteins, abundant nitrogen, and minerals for the growth of microorganisms [12]. Other than land application and biogas production [13], POME is not known for being used on food or other sectors. Using POME for biosurfactant production has several other advantages, including economics and reuse of industrial waste [14–16]. Therefore, POME has the potential for biosurfactant production for MEOR. This work examines the characteristics of biosurfactants potential for microbial enhanced oil recovery produced by *H meridiana* BK-AB4 with the POME substrate.

## 2. Materials and Methods

### 2.1. Microorganism

*H. meridiana* BK-AB4 was isolated from Bledug Kuwu, Central Java, a collection from the Bandung Institute of Technology Biochemistry Research Division [9]. Before being varied in the production optimization process, *H. meridiana* BK-AB4 was adapted to the POME carbon source. *H. meridiana* BK-AB4 was cultured in nutrient broth (NB) medium containing 5% NaCl (*w/v*). The culture was then incubated at 37 °C for 24 h while shaking at 150 rpm.

### 2.2. Media and Culture Conditions

*H. meridiana* BK-AB4 was maintained at 4 °C on nutrient agar (NA) medium in an agar slant, with 5% (*w/v*) NaCl added. One colony of bacterial cells from the medium in the agar slant was cultivated in NB medium at 37 °C with 150 rpm aeration for 24 h. For biosurfactant production, optimum results from RSM will be used.

## 2.3. Adaptation of H. meridiana BK-AB4 in POME

Before being varied in the production optimization process, the *H. meridiana* BK-AB4 was adapted to the POME carbon source. *H. meridiana* BK-AB4 were cultured in NB media containing 5% NaCl (*w/v*) and 2% POME (*v/v*). Next, the culture was incubated at 37 °C for 24 h while shaking 150 rpm.

## 2.4. Preparation and Characterization of Palm Oil Mill Effluent (POME)

The POME samples were placed in an oven at 80 °C for 1 d to allow the water in the POME to evaporate (dewatering process). The POME was then sterilized by autoclaving at 1 °C for 15 min to remove bacteria. Before GC-MS analysis, POME was converted into fatty acid methyl esters (FAMEs) through esterification reactions [17]. Each POME sample was placed in a round flask, and a methanol mixture of 10:1 was added. Sulfuric acid was as a catalyst (2.5% by weight of POME analyzed). The flask was connected to a reflux condenser. The mixture was heated to 100 °C using an electric heater for 1 h. The mixture was left to cool and then transferred to a separating funnel. Hot distilled water was added to the mixture, and extraction was repeated. The two layers formed were separated, and the top layer (i.e., methyl ester fatty acids) was removed. Extraction was stopped after the water phase became clear, and the pH reached 7. GC-MS characterized the composition of the methyl ester fatty acids.

## 2.5. Optimization of Biosurfactant Production

### 2.5.1. Single Factor Experiment (SFE)

A single factor analysis was carried out to find out the factors that influence the optimization of biosurfactant production. In addition, the purpose of this experiment was to determine the limit of the treatment interval as a factorial design box in response surface methodology. Test variables include POME concentration (*v/v*), incubation time (h), NaCl concentration and pH.

The fermentation culture was then centrifuged at 4000 rpm for 30 min to separate the bacterial cells. Subsequently, the supernatant was separated and used in the oil spreading test (OST). This technique was used aimed at measuring the diameter of the clear zone caused when a drop of a solution containing biosurfactant was placed on the oil-water surface [18]. A 14 cm diameter petri dish containing 40 mL distilled water was prepared for this measurement. One milliliter of oil was added to the surface of the water, and 10 µL of the supernatant from the optimization medium was then dropped on the oil surface. After 30 s, the diameter of the zone formed on the oil surface was measured called the oil displacement area (ODA) [18]. Larger diameter zones indicated better biosurfactant activity.

### 2.5.2. Response Surface Methodology (RSM)

The response surface methodology was applied to study the optimum condition for bacterial growth and produced biosurfactant using Design-Expert software, version 11.1.0.1 of StatEase. The variations in bacterial growth media conditions were the POME (*v/v*) concentration, incubation (h), NaCl (*w/v*) concentration, and pH. The test parameters used in this experiment are the same as the parameters used in SFE.

The optimization phase with RSM requires an experimental design to determine the number of experiments performed. Determination of optimum conditions was carried out with a matrix according to the rules of RSM with the center for four variables and one type of response. The variables tested were the POME (*v/v*) concentration, incubation (h), NaCl (*w/v*) concentration, and pH (X1, X2, X3, and X4 respectively). The response parameter was the ODA value (Y1). The combination of the amount of data taken in the optimization of this study was based on the CCD experimental design algorithm. The selection using CCD was based on the method most widely used for the optimization of several variables, and can also determine the equation of the regression model and optimal conditions in the experiment [19,20]. In this study, using four variables, and this combination resulted in 30 treatment points that had to be made with different X variable values.

*2.6. Biosurfactant Extraction*

Biosurfactant production was carried out by culturing *H. meridiana* BK-AB4 in 1 L of NB culture. The bacteria were incubated at 37 °C, with shaking at 150 rpm for 48 h. Biosurfactant extraction from the liquid culture was carried out by the Folch extraction method. The liquid bacterial culture was centrifuged at 4000 rpm for 30 min to separate the bacterial solids. The supernatant was then deposited by adding 6 M NaOH until the pH reached 12, and it was allowed to settle overnight in a refrigerator. The supernatant was then centrifuged again at 4000 rpm for 30 min to separate the solids. The suspension that formed at the base was removed, and chloroform:methanol (2:1) was added, followed by stirring. The formation of a faded yellow color in the chloroform–methanol phase indicated that the biosurfactant had been successfully extracted. For evaporation, a rotary evaporator was used until a crude extract of dry biosurfactant was obtained [21].

*2.7. Identification of the Biosurfactants Structure*

Fourier transform infrared spectroscopy (FT-IR) at most used to explain a few unknown mixture components. In addition, its use can identify the types of chemical bonds (functional group) [22]. The infrared spectrum of the biosurfactant extract was analyzed by FTIR using Shimadzu 500. The FTIR analysis was carried out at wavenumbers 4000–400 $cm^{-1}$. The peaks in the infrared spectrum pointed to a functional group that provided vibrational energy when given electromagnetic waves at certain wavenumbers [23]. The determination of functional groups in the biosurfactant can aid the identification of the type of biosurfactant produced. After the FTIR analysis, the infrared spectrum was interpreted based on the infrared spectrum table of Solomons and Fryhle [24].

The composition of the crude biosurfactant extract was also analyzed by using liquid chromatograph-mass spectrometer (LC-MS) Shimadzu Corps-MS/MS 3200 QTRAP ABSCIEX with a TSK-80s column (15 cm, 4.6 mm) and eluted using 80% (MeOH + 0.1% TFA): 20% ($H_2O$) for 30 min. Monitoring was performed at 365 nm, a temperature of 30 °C, a flow rate of 0.6 mL/min and a pressure of 150–160 bar. For the LC-MS analysis, the crude extract was first dissolved in methanol. The chromatogram and mass spectra produced were processed using Masslynx software.

*2.8. Surface Properties*

2.8.1. Surface Tension (ST) and Critical Micelle Concentration (CMC)

Surface tension was measured on a ring tensiometer (Krüss tensiometer K12) using the bacterial supernatant solution (50 mL) at 36 °C. The concentration at which micelles began to form was represented as the critical micelle concentration (CMC). The CMC was automatically determined by measuring the surface tensions of the purified biosurfactant in distilled water up to a constant value of surface tension [25].

2.8.2. Interfacial Tension (IFT)

Interfacial tension of biosurfactant was considered using Spinning Drop Tensiometer TX-500 C/D. Interfacial tension at different concentrations ranging from 100 to 500 mg/L of biosurfactant solutions was carried out and measured against light crude oil ($^0$API: 44.67) at 60 °C. In this study used formation water as control with measurements on different days to see the stability of work and durability of biosurfactant storage.

2.8.3. Emulsification Index ($E_{24}$)

Emulsification test is carried out to measure the ability from biosurfactant extracted to emulsify oil and water by looking at the stability of the emulsion formed after 24 h. $E_{24}$ was determined by adding 2 mL different hydrocarbons to 2 mL of cell-free supernatant, vortex at high speed for 5 min, and the

mixture was incubated at ambient temperature for 24 h. The $E_{24}$ index was given as a percentage of the height of the emulsified layer divided by the total height of the liquid column [26].

### 2.9. Enhanced Oil Recovery Experiment

Set Berea sandstone cores (diameter: 3.769 cm; length: 2.32 cm) with average permeability and porosity of 200–250 mD and 26.55%, respectively, were used for the imbibition test. The formation of water and crude oil in these experiments was obtained from the Indonesian oil field, which has an average reservoir temperature of 60 °C. The crude oils used for this test were of API 44.67°, with observations carried out periodically for 60 h.

## 3. Results and Discussion

### 3.1. Characterization of POME

In this study, POME was used as a carbon source substrate for *H. meridiana* BK-AB4 biosurfactant production. Conversion of the fatty acids to methyl esters was done to enable GC-MS analysis. POME was converted into a simple volatile derivative via an esterification reaction to form FAMEs. This esterification reaction was achieved using an acid catalyst ($H_2SO_4$) to form intermediate compounds. Figure 1 shows a common esterification reaction of fatty acid by acid catalyst [27].

**Figure 1.** Schematic reaction esterification of fatty acids by acid catalyst [27].

Based on the GC-MS chromatogram of the esterified POME sample (Table 1), the esterification reaction proceeded quite well and could be used as a basis for estimating the composition of the POME used. The compounds 9(E)-Octadecenoic acid methyl ester ($C_{19}H_{36}O_2$) and hexadecanoic acid methyl ester ($C_{17}H_{34}O_2$) were found as the most existing, i.e., 60.78% and 30.78% respectively. This means that the fatty acid compounds 9(E)-Octadecenoic acid (oleic acid, C18:1) and hexadecanoic acid (palmitic acid, C16:0) were the main constituents of the POME.

**Table 1.** Fatty acid methyl ester (FAME) composition by GC-MS from esterified palm oil mill effluent (POME) samples.

| Chemical Compound | Composition (%) | Chemical Formula | Retention Time (Minute) |
|---|---|---|---|
| 9(E)-Octadecenoic acid, methyl ester | 60.78 | $C_{19}H_{36}O_2$ | 19.467 |
| Hexadecanoic acid, methyl ester | 30.78 | $C_{17}H_{34}O_2$ | 13.361 |
| Octadecanoic acid, methyl ester | 1.95 | $C_{19}H_{38}O_2$ | 26.833 |
| Tetradecanoic acid, methyl ester | 1.85 | $C_{15}H_{30}O_2$ | 9.576 |
| Hexadecanoic acid, 14-methyl-, methyl ester | 1.03 | $C_{18}H36O_2$ | 32.529 |
| Heptadecanoic acid, methyl ester | 0.77 | $C18H_{36}O_2$ | 16.403 |
| Methylene Chloride | 0.34 | $CH_2Cl_2$ | 1.544 |
| Octadecanoic acid, 11-methyl-, methyl ester | 0.33 | $C_{20}H_{40}O_2$ | 41.378 |
| 11-Eicosenoic acid, methyl ester | 0.25 | $C_{21}H_{40}O_2$ | 30.866 |
| n-Hexadecanoic acid | 0.23 | $C_{16}H_{32}O_2$ | 18.090 |
| 9(Z),11(E),13(E)-Octadecatrienoic acid methyl ester | 0.22 | $C_{19}H_{32}O_2$ | 29.879 |
| 9(E)-Octadecenoic acid | 0.21 | $C_{18}H_{34}O_2$ | 27.809 |
| Dodecanoic acid, methyl ester | 0.18 | $C_{13}H_{26}O_2$ | 6.592 |
| Pentadecanoic acid, methyl ester | 0.14 | $C_{16}H_{32}O_2$ | 11.270 |

The type of carbon source influences the nature of the biosurfactant produced. Previous studies used olive oil and palm oil as the carbon source [28]. POME, which was used in the present study contains both saturated and unsaturated fatty acids. Therefore, its components resemble those of palm oil.

Although POME is hazardous waste for the environment, the contents of organic materials such as carbohydrates, nitrogen compounds, fats, and minerals can act as a carbon source [29]. Microorganisms use these nutrients in growth and produce secondary metabolites such as biosurfactants by using specific enzymatic pathways so that they can utilize lipids in POME into biosurfactants [30].

Several studies comparing the use of several alternative carbon sources show that POME can help microorganisms produce good biosurfactants in reducing surface tension and emulsification [31]. The economic value of POME can solve the problem of the high cost of applying EOR and MEOR methods. As a carbon source substrate, POME is needed in biosurfactant production as an alternative raw material. The use of POME requires very low costs or can be obtained for free because it is a palm oil waste [32].

### 3.2. Optimization of Biosurfactant by Response Surface Methodology

A preliminary test was carried out before designing the experimental design for optimization. The single-factor analysis was used in this method to determine the impact of two or more factors on the response. The aim was to assess the important effects of treatment parameters on the formation of ODA values. In addition to knowing the appropriate range in optimization studies with RSM.

The experimental results shown in Figure 2 show the optimum range of treatment parameters, POME concentration, incubation time, NaCl concentration, and pH for ODA values are 10–25%, 100–140 h, 3–7%, and 6–8. In the treatment variable, pome concentration significantly decreased ODA value at a concentration of 30%. A similar thing happened: the ODA value decreased sharply at a concentration of 9% NaCl. POME, as a carbon source, plays an important role in the growth and production of biosurfactants. Desai and Banat [33] stated that the concentration of the carbon source used greatly determines the biosurfactant yield, and the type of carbon source used affects the composition of the biosurfactant produced. Meanwhile, on the observation of pH and incubation period, a significant decrease in ODA value did not occur. This was because, in the incubation range above 140 h, biosurfactants are still formed, but not as well as in the range 100–140 h. Likewise, with pH above 8, some *Halomonas* genera can still live at that pH, so biosurfactants can still be produced in this condition [34].

The results of the RSM approach to determine the optimum conditions for biosurfactant production by varying the POME concentration, incubation time, NaCl concentration, and pH are presented in Table 2 below.

**Table 2.** Combination treatment matrix for biosurfactant production optimization and value for variables of the RSM experimental design.

| Std | Run | Experimental Variable | | | | Oil Displacement Area (cm) | | Residual |
|-----|-----|-------|-------|-------|-------|--------------|-----------------|----------|
|     |     | $X_1$ | $X_2$ | $X_3$ | $X_4$ | Actual Value | Predicted Value |          |
| 30  | 1   | 17.5  | 120   | 5     | 7     | 4            | 3.565           | 0.435    |
| 15  | 2   | 10    | 140   | 7     | 8     | 2.4          | 1.968           | 0.432    |
| 20  | 3   | 17.5  | 160   | 5     | 7     | 1.8          | 2.534           | −0.734   |
| 27  | 4   | 17.5  | 120   | 5     | 7     | 3.5          | 3.565           | −0.065   |
| 8   | 5   | 25    | 140   | 7     | 6     | 3            | 2.885           | 0.115    |
| 3   | 6   | 10    | 140   | 3     | 6     | 2.2          | 2.168           | 0.032    |
| 5   | 7   | 10    | 100   | 7     | 6     | 2.8          | 2.703           | 0.097    |
| 21  | 8   | 17.5  | 120   | 1     | 7     | 2            | 2.402           | −0.402   |
| 29  | 9   | 17.5  | 120   | 5     | 7     | 3.4          | 3.565           | −0.165   |
| 4   | 10  | 25    | 140   | 3     | 6     | 2.8          | 2.339           | 0.461    |
| 14  | 11  | 25    | 100   | 7     | 8     | 2            | 1.653           | 0.347    |

**Table 2.** *Cont.*

| Std | Run | Experimental Variable | | | | Oil Displacement Area (cm) | | Residual |
| --- | --- | --- | --- | --- | --- | --- | --- | --- |
| | | $X_1$ | $X_2$ | $X_3$ | $X_4$ | Actual Value | Predicted Value | |
| 11 | 12 | 10 | 140 | 3 | 8 | 2.7 | 2.322 | 0.378 |
| 12 | 13 | 25 | 140 | 3 | 8 | 2.5 | 2.218 | 0.282 |
| 9 | 14 | 10 | 100 | 3 | 8 | 3.2 | 2.936 | 0.264 |
| 2 | 15 | 25 | 100 | 3 | 6 | 2.6 | 2.653 | −0.053 |
| 1 | 16 | 10 | 100 | 3 | 6 | 3.4 | 3.107 | 0.293 |
| 13 | 17 | 10 | 100 | 7 | 8 | 1.8 | 2.007 | −0.207 |
| 16 | 18 | 25 | 140 | 7 | 8 | 2.2 | 2.239 | −0.039 |
| 7 | 19 | 10 | 140 | 7 | 6 | 2.8 | 2.339 | 0.461 |
| 23 | 20 | 17.5 | 120 | 5 | 5 | 2.8 | 3.119 | −0.319 |
| 25 | 21 | 17.5 | 120 | 5 | 7 | 3.5 | 3.565 | −0.065 |
| 19 | 22 | 17.5 | 80 | 5 | 7 | 3 | 2.887 | 0.113 |
| 26 | 23 | 17.5 | 120 | 5 | 7 | 3.8 | 3.565 | 0.235 |
| 22 | 24 | 17.5 | 120 | 9 | 7 | 1.8 | 2.019 | −0.219 |
| 28 | 25 | 17.5 | 120 | 5 | 7 | 3.2 | 3.565 | −0.365 |
| 24 | 26 | 17.5 | 120 | 5 | 9 | 2 | 2.302 | −0.302 |
| 10 | 27 | 25 | 100 | 3 | 8 | 2 | 2.207 | −0.207 |
| 18 | 28 | 32.5 | 120 | 5 | 7 | 1.8 | 1.869 | −0.069 |
| 6 | 29 | 25 | 100 | 7 | 6 | 2.5 | 2.624 | −0.124 |
| 17 | 30 | 2.5 | 120 | 5 | 7 | 1.5 | 2.052 | −0.552 |

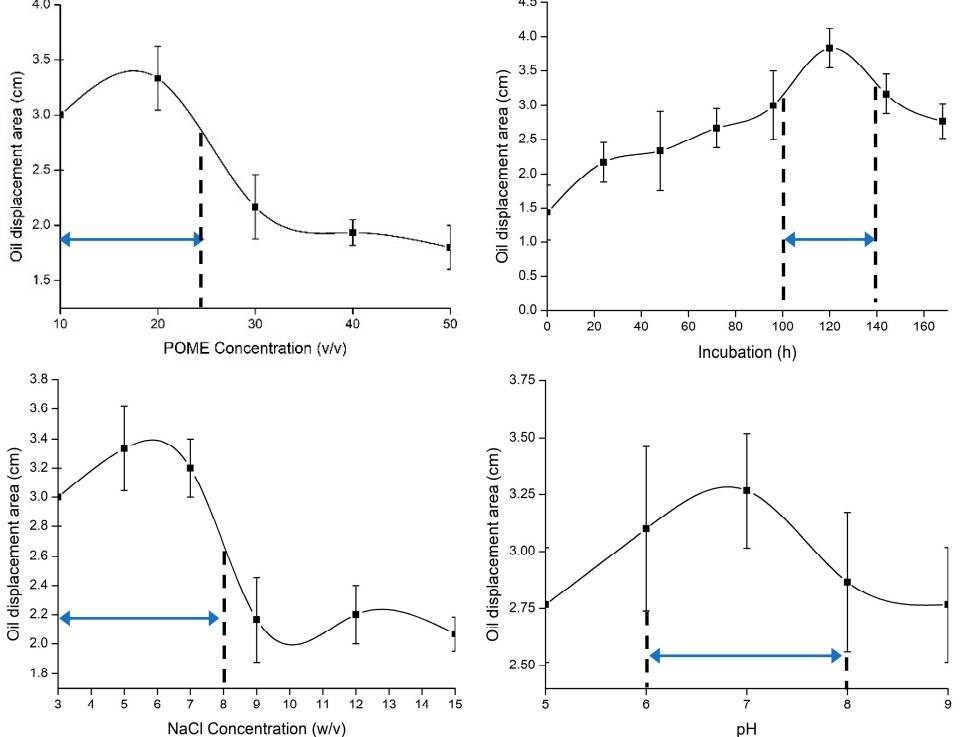

**Figure 2.** The range of optimization area (arrow) to the factors that influence the biosurfactant production.

Examination of the oil spreading test by measuring the oil displacement area indicates that the bacteria has produced biosurfactants [35]. The relationship between ODA as the dependent variable in each experimental condition, as stated in the Equation (1).

$$Y = -11.69968 + 0.151343\,(X_1) + 0.059115\,(X_2) + 0.717188\,(X_3) + 2.78646\,(X_4) + 0.001042\,(X_1)(X_2)$$
$$+ 0.006250\,(X_1)(X_3) - 0.009167\,(X_1)(X_4) + 0.003594\,(X_2)(X_3) + 0.004062\,(X_2)(X_4)$$
$$- 0.065625\,(X_3)(X_4) - 0.007130\,(X_1)^2 - 0.000534\,(X_2)^2 - 0.084635\,(X_3)^2$$

$$(1)$$

where Y = ODA (cm) as dependent variable. The independent variables were X1 = % (*v/v*) POME, X2 = incubation time (h), X3 = % (*w/v*) NaCl, X4 = pH. The regression coefficient: $-11.69968$ is a constant coefficient; 0.151343, 0.059115, 0.717188, and 2.78646 are the linear coefficients; 0.001042, 0.006250, 0.009167, 0.003594, 0.004062, and 0.065625 are the interaction coefficient between variable X1, X2, X3, X4, and 0.007130, 0.000534, and 0.084635 are the quadratic coefficients.

When processing data with Design-Expert software version 11.1.0.1. Optimization tests were performed with several mathematical equation models, from the results of this test, the quadratic model was chosen because it produced statistical variables that showed the best desirability. Based on the ANOVA test, obtained the coefficient of determination $R^2$ of 0.8068, which indicates that 80.68% of the sample variables in biosurfactant production was influenced by the independent variables. Likewise, the *p*-value was 0.0081 (<0.005), which means the model was significantly following the observed variations (Table 3). Therefore, this regression model can be satisfactorily accepted and useful to get the best ODA value. Determination of the optimum conditions for biosurfactant production was carried out by maximizing ODA response in Design-Expert software version 11.1.0.1. Optimum condition values are indicated by using POME 16% (*v/v*), incubation 112 h, NaCl 4.7% (*w/v*) and pH 6.7. The ODA value obtained was 3.6 cm, with the desirability of 0.857. Validation was performed with five repetitions to obtain an average ODA of 3.54 cm.

**Table 3.** ANOVA oil displacement area model test at response surface methodology (RSM).

| Source | Sum of Square | Mean of Square | *F*-Value | *p*-Value | |
|---|---|---|---|---|---|
| **Model** | 10.15 | 0.7251 | 3.73 | 0.0081 | significant |
| $X_1$-POME | 0.0504 | 0.0504 | 0.2594 | 0.6180 | |
| $X_2$-Incubation | 0.1838 | 0.1838 | 0.9453 | 0.3463 | |
| $X_3$-Salinity | 0.2204 | 0.2204 | 1.13 | 0.3038 | |
| $X_4$-pH | 1.00 | 1.00 | 5.15 | 0.0385 | |
| $X_1 X_2$ | 0.3906 | 0.3906 | 2.01 | 0.1768 | |
| $X_1 X_3$ | 0.1406 | 0.1406 | 0.7234 | 0.4084 | |
| $X_1 X_4$ | 0.0756 | 0.0756 | 0.3890 | 0.5422 | |
| $X_2 X_3$ | 0.3306 | 0.3306 | 1.70 | 0.2118 | |
| $X_2 X_4$ | 0.1056 | 0.1056 | 0.5434 | 0.4724 | |
| $X_3 X_4$ | 0.2756 | 0.2756 | 1.42 | 0.2523 | |
| $X_1^2$ | 4.41 | 4.41 | 22.69 | 0.0003 | |
| $X_2^2$ | 1.25 | 1.25 | 6.43 | 0.0228 | |
| $X_3^2$ | 3.14 | 3.14 | 16.17 | 0.0011 | |
| $X_4^2$ | 1.25 | 1.25 | 6.43 | 0.0228 | |
| **Residual** | 2.92 | 0.1944 | | | |
| Lack of Fit | 2.50 | 0.2502 | 3.03 | 0.1166 | not significant |
| Pure Error | 0.4133 | 0.0827 | | | |
| **Cor Total** | 13.07 | | | | |

Table 2 shows successive correlations between the actual value and predicted value for biosurfactant yields observed through ODA values. Small residual value in each run of the four variants, the model is statistically significant. This good result was also supported by good handling in the production process. The accuracy of measurement and weighing becomes important to support this result. The predicted value is an estimated value obtained from the selection of mathematical equation models when inputting data in Design-Expert software version 11.1.0.1. Estimated values in the predicted value column indicate the extent of bias that occurs in the experimental data that was owned. These values will correlate and affect all statistical parameters produced. The smaller the difference in value between the actual value and predicted value, the better the value of the statistical parameters obtained.

Further validation of independent variables interaction was performed by visualizing three-dimensional is shown in Figure 3. The three-dimensional surface plot response plays a very important role in the surface response analysis study. For surface responses using second-order regression equations, a plot like the one above can be more complex than the simple parallel series



plot that can occur on first-order plot models. Three-dimensional surface plot responses indicate the possible independence of factors with responses. The red dots on the three-dimensional surface plot response indicate the optimum area on the surface response, the optimum point obtained at the top of the red area.

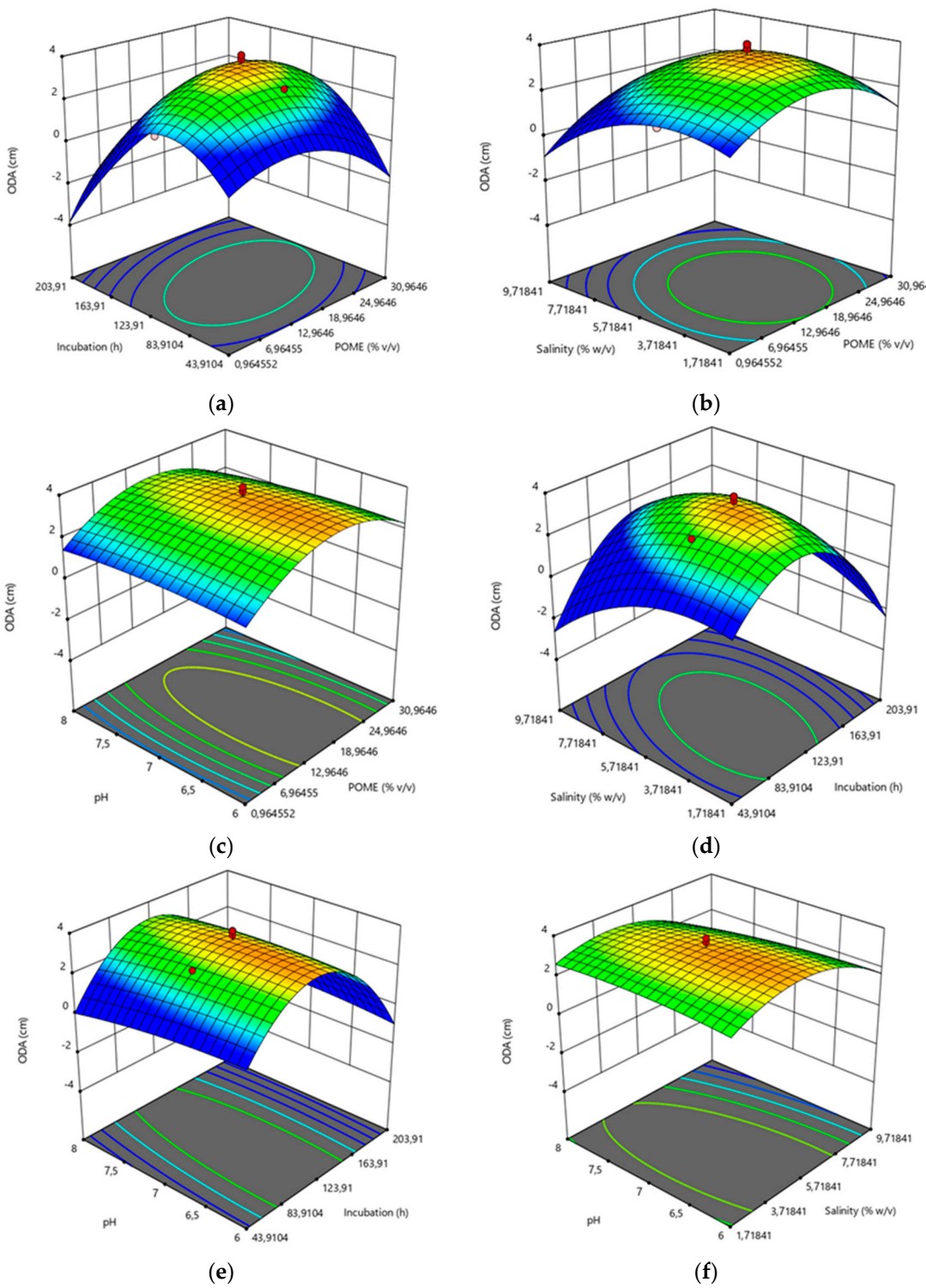

**Figure 3.** Intervariable relationship in RSM oil displacement area response. (**a**) Incubation and POME concentration, (**b**) NaCl concentration and POME concentration, (**c**) pH and POME concentration, (**d**) NaCl concentration and incubation, (**e**) pH and incubation, (**f**) pH and NaCl concentration.

### 3.3. Structural Characterization

### 3.3.1. FTIR Analysis of Biosurfactant Molecule

The extracted biosurfactant from the infrared spectrum was analyzed by FTIR at wavenumbers 4000–400 cm$^{-1}$ to determine the functional groups possessed by biosurfactants. The detected functional groups showed that the crude biosurfactant extract produced was a class of fatty acids (Figure 4).

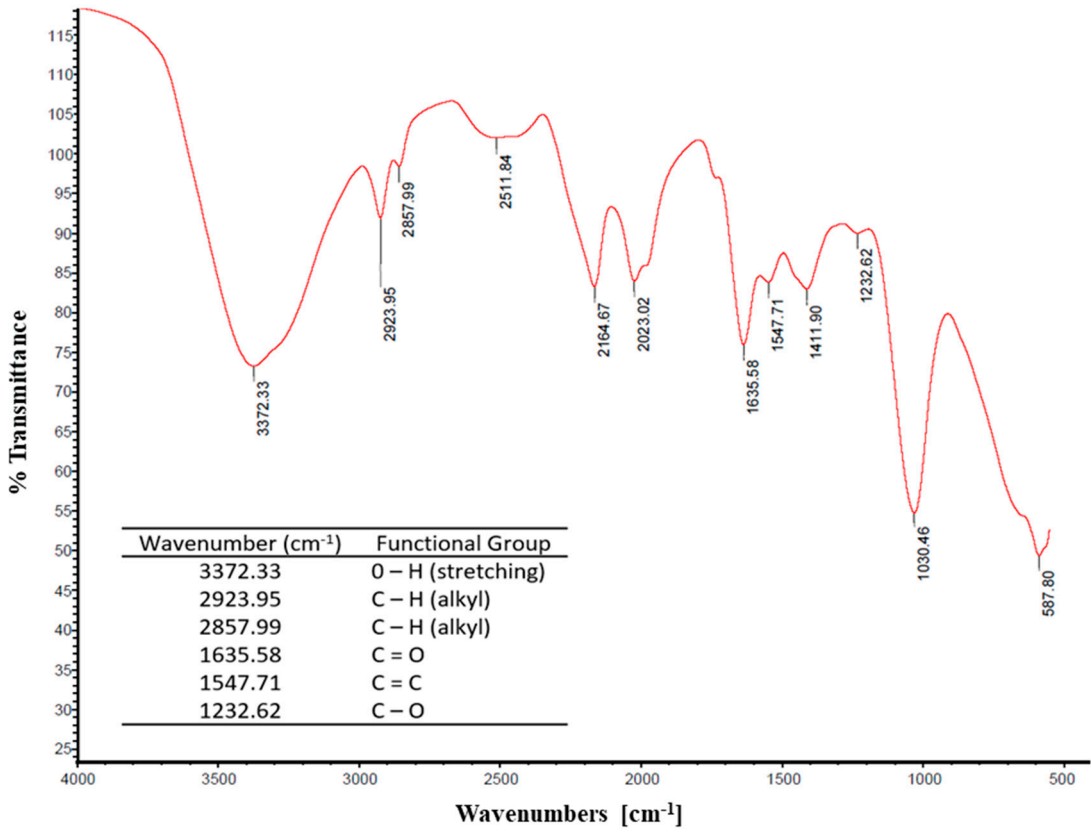

| Wavenumber (cm$^{-1}$) | Functional Group |
| --- | --- |
| 3372.33 | O − H (stretching) |
| 2923.95 | C − H (alkyl) |
| 2857.99 | C − H (alkyl) |
| 1635.58 | C = O |
| 1547.71 | C = C |
| 1232.62 | C − O |

**Figure 4.** FT-IR spectrum analysis of biosurfactants *Halomonas meridiana* BK-AB4.

The existence of bonds of O-H at 3372.33, C=O at 1635.58, and C-O at 1232.62 denotes the presence of functional groups of carboxylic acids or esters in biosurfactant samples [24]. The detection of a carboxylic acid functional group indicates that the biosurfactant produced belongs to the fatty acid class of biosurfactants, whereas the presence of C-H bonds denotes the existence of long hydrocarbon chains composed of hydrophobic or non-polar groups typically found in biosurfactants [36]. A C=C bond points to the presence of a double bond, which may be influenced by unsaturated fatty acids used as carbon sources [3,37]. According to the functional groups detected in the FTIR analysis, the crude biosurfactant extract produced belonged to the fatty acid class of biosurfactants. The results obtained were similar to those reported by Sari et al. [3].

### 3.3.2. LC-MS Analysis of Biosurfactant Molecule

For the liquid chromatography-mass spectrometry (LC-MS) analysis, the biosurfactant crude extract was first dissolved into methanol before injection to the chromatography column. Figure 5 shows the chromatogram obtained from the LC-MS analysis. The mass spectrum of the chromatogram peak area was analyzed to enable a quantitative analysis of the crude biosurfactant extract and identify the composition percent (% w). Four high peaks were detected at different retention times.

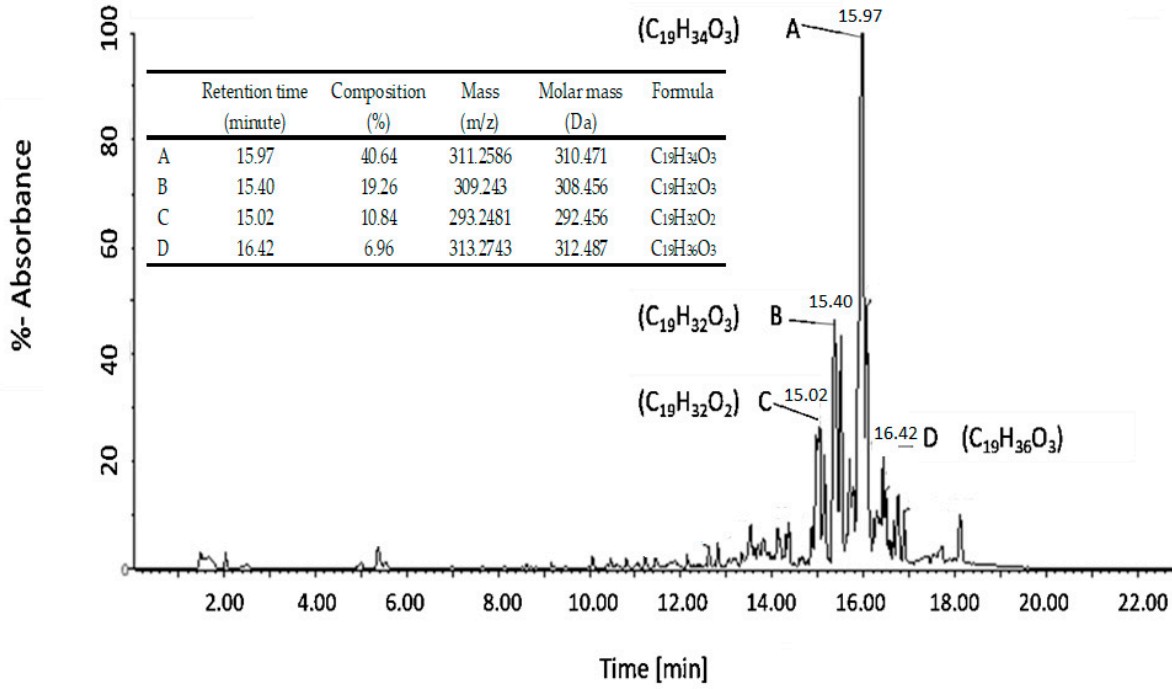

**Figure 5.** LC-MS analysis of biosurfactant from *H. Meridiana* BK-AB4.

In Figure 6 the main compounds are 9(E), 11(Z)-Octadecadienoic acid, 13-hydroxy-, methyl ester ($C_{19}H_{34}O_3$); 9(E), 11(E)-Octadecadienoic acid, 13-oxo-, methyl ester ($C_{19}H_{32}O_2$); 6,9,12-Octadecatrienoic acid, methyl ester ($C_{19}H_{32}O_2$); and 3-Oxononadecanoic acid ($C_{19}H_{36}O_3$). Those four compounds consist of 77.7% of the biosurfactant produced.

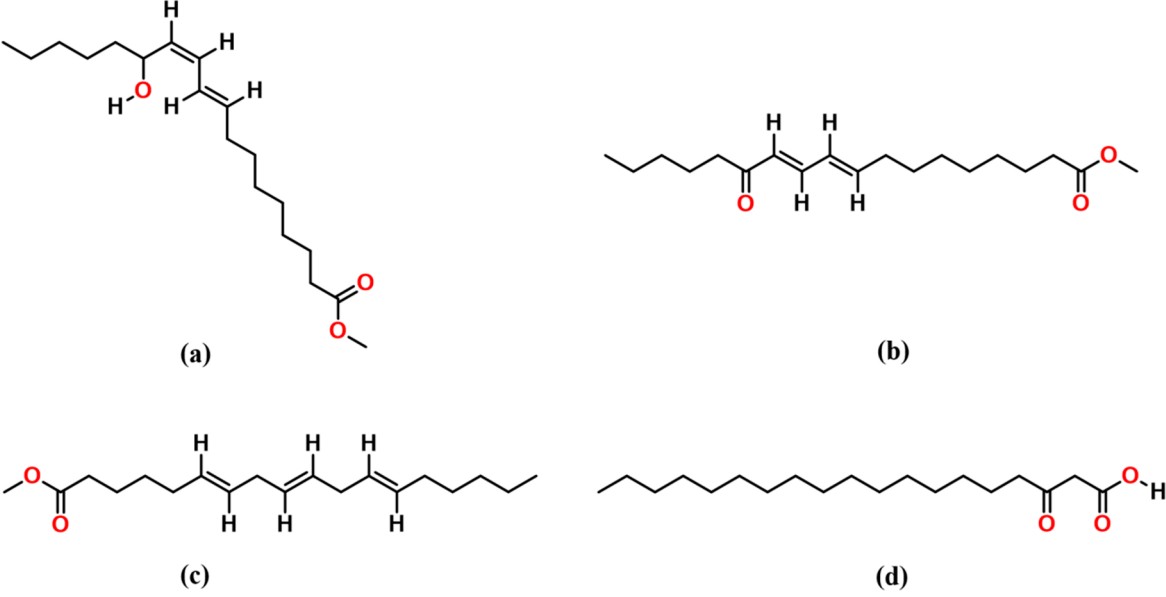

**Figure 6.** Structure of biosurfactant production of *H. meridiana* BK-AB4, (**a**) 9(E),11(Z)-Octadecadienoic acid, 13-hydroxy-, methyl ester, (**b**) 9(E),11(E)-Octadecadienoic acid, 13-oxo-, methyl ester, (**c**) 6,9,12-Octadecatrienoic acid, methyl ester, and (**d**) 3-Oxononadecanoic acid.

In LC-MS measurements, the detector analyzes the substance based on retention time based on peak intensity and peak area. Four detected compounds as previously have a polar group at the end, with a carboxyl (–COOR') group that is soluble in water, as well as non-polar groups in the

form of hydrocarbon chains, which are soluble in organic solvents and dissolve the oil (non-polar). Thus, they can be classified as fatty acid biosurfactants (Figure 6).

Carboxyl (–COOR') functional groups are polar (hydrophilic) molecules that dissolve in water. Alkyl groups with C-H bonds indicate a hydrocarbon chain that acts as a non-polar or hydrophobic group, which is soluble only in organic solvents and can dissolve oil, which is also non-polar. As compounds with dual solubility can form emulsions when mixed with water, they are good emulsification agents [36]. Fatty acid biosurfactants can be a mixture of carboxylic acids and neutral lipids, such as esters, alcohols, and glycerides [38]. In addition to the production of straight-chain fatty acids, microorganisms can produce complex fatty acids with alkyl branches and hydroxyl (-OH) groups [36,39]. Three of the four structures detected in the LC-MS analysis were methyl esters of fatty acids but contained similar polar and non-polar groups. The results of the LC-MS analysis validated the provisional results of the FTIR analysis, namely the presence of carboxylic acid groups in fatty acids, hydrocarbon chains as non-polar biosurfactants, alkyl in methyl esters, and double carbon bonds.

### 3.4. Analysis of Physico-Chemical Properties of Biosurfactant

Efficiency is an important character of good biosurfactants that can be measured with CMC. One of the characteristics of biosurfactants is the ability to reduce surface-water stress. The concentration value (CMC) is determined by drawing a straight line between the tension surface drop points and the points when the addition of surfactants no longer decreases the surface tension [40]. In this measurement, purified biosurfactants are dissolved in distilled water. Then surface tension was measured with various biosurfactant concentrations. Based on Figure 7, it can be concluded that the reduction in surface tension of water was 54.16 mN/m, and the CMC value was 350 mg/L. In its application in industry, the value of CMC was useful to determine the use of biosurfactants in certain fields [40]. The CMC value for surfactants applied in MEOR was generally in the range of 1–2000 mg/L, and good surfactants can reduce water surface tension from 72 to 35 mN/m [41].

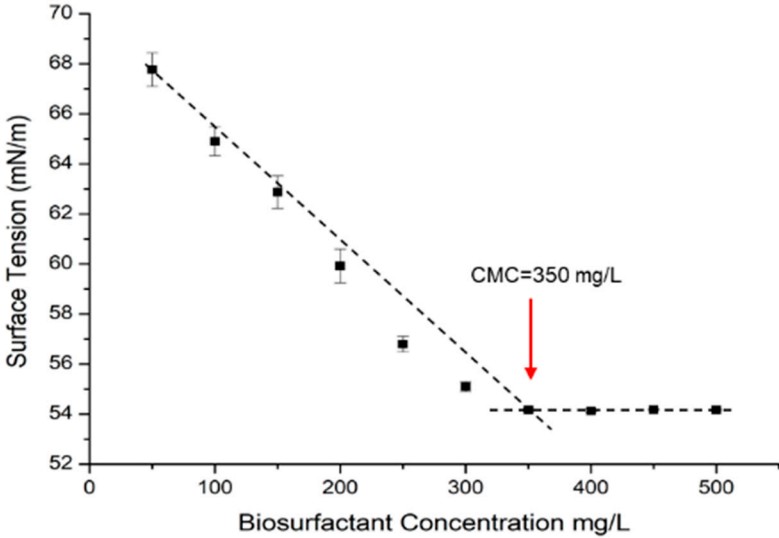

**Figure 7.** Critical micelle concentration (CMC) and minimum surface tension reduction by crude biosurfactant produced by *H. meridiana* BK-AB4 at 36 °C.

In the measurement of interfacial tension, the value obtained was due to the molecular force interface between two fluids. Furthermore, IFT will unite these molecules. The potential biosurfactants will work by entering into the interfaces of two liquids or a liquid and a gas, so it can reduce the interfacial tension between water and the hydrophobic substance (crude oil) tested as shown in Figure 8. On the second and seventh day observations, significant IFT values occurred at a biosurfactant concentration of 300 mg/L, respectively, 0.17 and 1.27 m/m. While the lowest IFT value is achieved on

the second and seventh-day calculations at a concentration of 500 mg/L, i.e., 0.03 mN/m and 0.06 mN/m, respectively. On the 14th day until the 20th day, the biosurfactant working capacity began to rise, as seen from the IFT values produced at various biosurfactant concentrations.

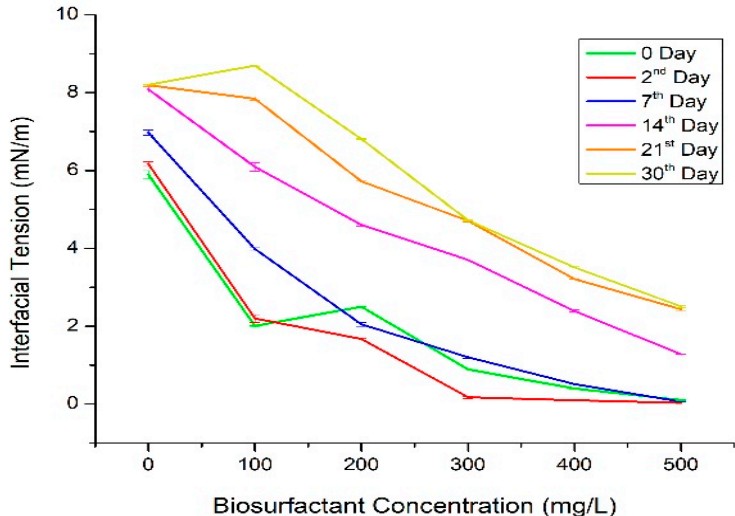

**Figure 8.** Measurement of interfacial tension at different biosurfactant concentrations in water at different temperatures.

Lowering this power to a low value can significantly improve oil recovery. This occurs because IFT gives rise to capillary forces in the porous media, which are mainly responsible for the hydrocarbons trapped in the form of residual saturation. The results show that biosurfactants are very effective. Therefore, the efficiency of biosurfactants in reducing interfacial tension water and hydrophobic substances makes it more attractive for use in enhanced oil recovery (EOR) [42].

Biosurfactant activity was carried out by calculating the emulsification index value of palm oil biosurfactants after 24 h ($E_{24}$). The emulsification index is a quantitative parameter to determine the ability of biosurfactants to form emulsions between two compounds with different polarities. Biosurfactants are used to mix the two compounds with the hydrophilic part of the biosurfactant will interact with polar compounds, while the hydrophobic part will interact with compounds that have low polarity so that an emulsion was formed. A good biosurfactant emulsifying property is its potential to instigate oil-water emulsification [43]. The concentration used to determine the biosurfactant emulsification activity of *H. Meridiana* BK-AB4 was CMC concentration (300 mg/L). Based on the calculation of the value of $E_{24}$, the highest percentage was obtained in crude oil CR-04. This is directly proportional to the IFT value with the lowest value of 0.18 mN/m (Table 4). According to De et al. [44]. The size of the $E_{24}$ value is strongly influenced by the molecular mass of biosurfactants.

**Table 4.** Biosurfactant emulsification activity and interfacial tension (IFT) value of *H. meridiana* BK-AB4.

| Crude Oil | Classification | $^0$API | E24 (%) | IFT (mN/m) |
|---|---|---|---|---|
| CR-01 | Intermediate–paraffinic | 39.1 | 50 | 1.56 |
| CR-02 | Intermediate–intermediate | 32.7 | 63 | 0.72 |
| CR-03 | Paraffinic–intermediate | 26.0 | 53 | 1.67 |
| CR-04 | Naphthenic–naphthenic | 38.7 | 76 | 0.18 |

*3.5. Imbibition Test Result*

Imbibition test was conducted to determine the performance of surfactants in lifting oil from synthetic rocks. The saturated cores are dried then weighed. It is intended that the pores of rocks made can be easily entered by oil. Biosurfactant performance can be determined based on recovery

oil, or the amount of crude oil raised by the biosurfactant from the total crude oil stored in the core. In other words, the volume of oil raised from the rock was calculated as a percent of the amount of saturated oil. The greater the volume of oil that can be raised, it means that the ability of biosurfactants to reduce the interface tension between oil and water was getting better. Details of the imbibition test, along with results for 60 h, are shown in Figure 9. The oil recovery increases with the time of soaking the cores in the biosurfactant solution. The average percentage increase in oil recovery or other words as original oil in place (% OOIP) per day ranges from 1–3%. The highest increase occurred at the 18th hour of observation, which was 2.72%.

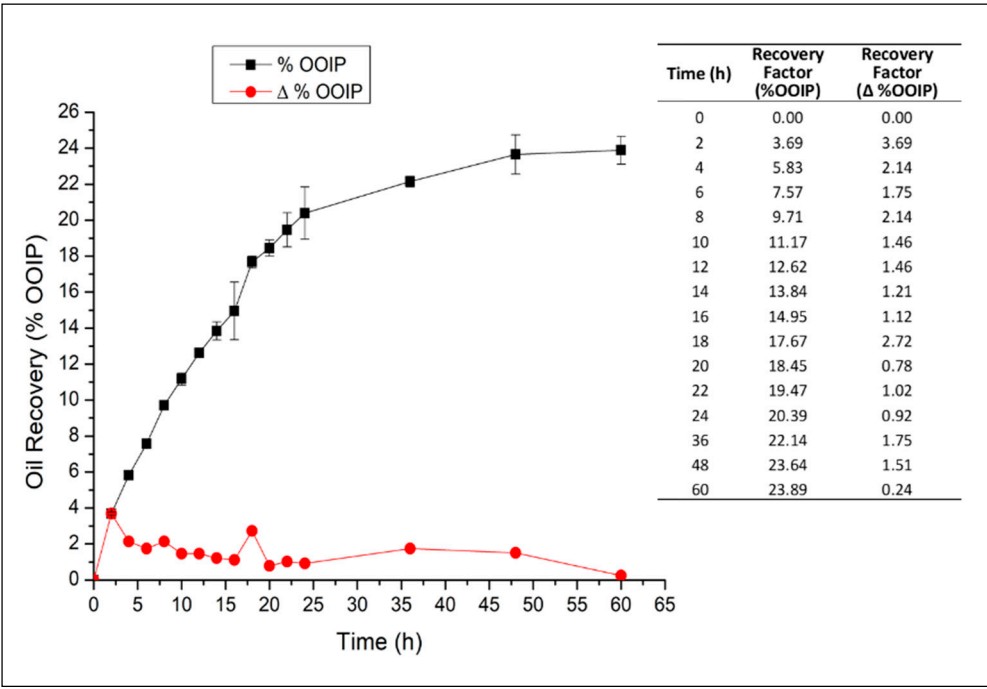

**Figure 9.** Residual oil recovery by applying the biosurfactant solution from *H. meridiana* BK-AB4.

In general, the results obtained from this study are following the results that have been reported in previous studies. A study by Khajepour et al. [45] reported that there was 4.4% of extra oil recovered after biosurfactant injection produced by strains of *Enterobacter cloacae* in a glass micromodel system. Recovery of 6% oil using biosurfactants produced by *Bacillus mojavensis* (PTCC 1696) on low permeability of dolomite cores [46]. Câmara et al. [47] used biosurfactant from *Pseudomonas aeruginosa,* which resulted in oil recovery of 3–7% and 7–11% during the recovery trial using a synthetic sandstone plug.

## 4. Conclusions

Based on the optimization results using response surface methodology, the biosurfactant was optimally produced after fermentation for 112 h of in POME concentration of 16% (*v/v*), NaCl concentration of 4.7 (*w/v*), and pH 6.5, with an oil displacement area of 3.642 cm. The produced biosurfactant has high emulsification activity in light crude oil ($E_{24}$ value 76%) with the interfacial tension value is 0.18 mN/m and ability to lower the surface tension from 67.76 to 54.16 mN/m with an effective CMC of 350 mg/L. Further treatment with imbibition experiment results 23.89% additional oil recovery factor. Therefore, with all the parameter above, the halophilic biosurfactant-producing microorganism that was isolated from the Bledug Kuwu mud volcano in Central Java Indonesia could be implemented for MEOR application in the petroleum industry.

**Author Contributions:** C.N.S. performed the design of the study, analyzed the data, and wrote the manuscript. R.H. supervised the lab work and the study. A.F.P.H. performed graphical works. M.Y.A.R. performed the project

administration. M.G. supervised the design of the research and organized the writing. All authors have read and approved the final manuscript.

**Funding:** We gratefully acknowledge the research grant from the Ministry of Research Technology and Higher Education through PDUPT (NKB-1646/UN2.R3.1/HKP.00/2019) and publication grant from Universitas Indonesia through PUTI KI (NKB-784/UN2.RST/HKP.05.00/2020).

**Conflicts of Interest:** The authors declare no conflict of interest.

## Abbreviations

| Abbreviation | Full Name |
| --- | --- |
| API | American Petroleum Institute |
| CCD | Central composite design |
| CMC | Critical micelle concentration |
| $E_{24}$ | Emulsification Index |
| EOR | Enhanced oil recovery |
| FAMEs | Fatty acid methyl esters |
| IFT | Interfacial tension |
| MEOR | Microbial enhanced oil recovery |
| ODA | Oil displacement area |
| OOIP | Original oil in place |
| OST | Oil spreading test |
| POME | Palm oil mill effluent |
| RSM | Response surface methodology |

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
