# Peer review of "Process Optimization of Palm Oil Mill Effluent-Based Biosurfactant of Halomonas meridiana BK-AB4 Originated from Bledug Kuwu Mud Volcano in Central Java for Microbial Enhanced Oil Recovery"

_processes, doi:10.3390/pr8060716_

Round 1

Reviewer 1 Report

Review of the article titled ‘Process Optimization of Palm Oil Mill Effluent-Based 3 Biosurfactant of Halomonas meridiana BK-AB4 4 Originated from Bledug Kuwu Mud Volcano in 5 Central Java for Microbial Enhanced Oil Recovery’. The manuscript is well written. The results are comprehensive and will be very useful for others researches so I recommend the present manuscript to publication in Processes after minor revision.

Point 2.7 - Please provide specific LC-MS identification parameters.

In which industries can Halomonas meridiana BK-AB4 surfactants be used?

Author Response

The revisions can be seen in "Track Changes" function in Microsoft Words file of the manuscript. Line number is based on "final" display in "Track Changes" function. 

Reviewer 2 Report

In this manuscript, the authors present detailed optimization of the use of biosurfactants produced from H. Meridiana for oil recovery. They calibrated the conditions required such as pH, incubation time, and others for getting good process. The work is well performed, I have only few minor comments.

Some of the figures suffer from low resolution.

The conclusion section should be written to better understand what are the preferred conditions.

The calculated CMC was measured at what temperature? can it be stated in the caption?

An abbreviation table should be added - it is really hard to follow the manuscript without it.

Author Response

The revisions can be seen in "Track Changes" function in Microsoft Words file of the manuscript. Line number is based on "final" display in "Track Changes" function.

The revisions for Figure 4-6 are yellow-highlighted directly in the manuscript (not visible in Track Changes). 
